# Skin Phototype Could Be a Risk Factor for Multiple Sclerosis

**DOI:** 10.3390/jcm9082384

**Published:** 2020-07-26

**Authors:** Patricia Urbaneja, Isaac Hurtado-Guerrero, Miguel Ángel Hernández, Begoña Oliver-Martos, Celia Oreja-Guevara, Jesús Ortega-Pinazo, Ana Alonso, Francisco J Barón-López, Laura Leyva, Óscar Fernández, María Jesús Pinto-Medel

**Affiliations:** 1Instituto de Investigación Biomédica de Málaga-IBIMA, 29009 Málaga, Spain; urbanejaromero@gmail.com (P.U.); ishugu@gmail.com (I.H.-G.); begoliver@gmail.com (B.O.-M.); jesusortegapinazo@gmail.com (J.O.-P.); anaat73@hotmail.com (A.A.); baron@uma.es (F.J.B.-L.); leyvafer@gmail.com (L.L.); 2UGC Neurociencias, Hospital Regional Universitario de Málaga, 29010 Málaga, Spain; 3Red Temática de Investigación Cooperativa: Red Española de Esclerosis Múltiple REEM (RD16/0015/0010), 28049 Madrid, Spain; mhernandezp78@hotmail.com (M.Á.H.); orejacbn@gmail.com (C.O.-G.); 4Unit of Multiple Sclerosis, Department of Neurology, Hospital Universitario Ntra. Sra. de Candelaria, 38010 Santa Cruz de Tenerife, Spain; 5Department of Neurology, Hospital Clínico San Carlos, Departamento de Medicina, Facultad de Medicina, Universidad Complutense de Madrid (UCM), IdISSC, 28040 Madrid, Spain; 6Unit of Biostatistics, Deparment of Public Health, Faculty of Medicine, University of Malaga, 29010 Málaga, Spain; 7Department of Nursing, Faculty of Health Sciences, University of Malaga, 29010 Málaga, Spain; 8Department of Pharmacology, Faculty of Medicine, University of Malaga, 29010 Málaga, Spain

**Keywords:** multiple sclerosis, skin phototypes, smoking, vitamin D, HLA

## Abstract

Environmental and genetic factors are assumed to be necessary for the development of multiple sclerosis (MS), however its interactions are still unclear. For this reason here, we have not only analyzed the impact on increased risk of MS of the best known factors (*HLA-DRB1*15:01* allele, sun exposure, vitamin D levels, smoking habit), but we have included another factor (skin phototype) that has not been analyzed in depth until now. This study included 149 MS patients and 147 controls. A multivariate logistic regression (LR) model was carried out to determine the impact of each of the factors on the increased risk of MS. Receiver Operating Characteristics (ROC) analysis was performed to evaluate predictive value of the models. Our multifactorial LR model of susceptibility showed that females with light brown skin (LBS), smokers and who had *HLA-DRB1*15:01* allele had a higher MS risk (LBS: OR = 5.90, IC95% = 2.39–15.45; smoker: OR = 4.52, IC95% = 2.69–7.72; presence of *HLA-DRB1*15:01*: OR = 2.39, IC95% = 1.30–4.50; female: OR = 1.88, IC95% = 1.08–3.30). This model had an acceptable discriminant value with an Area Under a Curve AUC of 0.76 (0.69–0.82). Our study indicates that MS risk is determined by complex interactions between sex, environmental factors, and genotype where the milieu could provide the enabling proinflammatory environment that drives an autoimmune attack against myelin by self-reactive lymphocytes.

## 1. Introduction

Multiple sclerosis (MS) is an inflammatory, demyelinating and neurodegenerative autoimmune disease of the central nervous system (CNS), most probably originated by a complex interaction of genetic and environmental factors [1]. The genetic component of MS is the result of the action of common allelic variants in several genes, being the human leukocyte antigen (HLA) gene cluster the strongest susceptibility locus. The HLA class II alleles, especially the haplotype *DRB5*01:01-HLA-DRB1*15:01-HLA-DQA1*01:02-HLA-DQB1*06:02* confers 2–4-fold increased risk of MS [2]. The International Multiple Sclerosis Genetics Consortium also identified 233 MS risk variants (single nucleotide polymorphisms or SNP’s), 200 outside the HLA complex [3]. Additionally, MS has been associated with several environmental factors such as high latitude, vitamin D deficiency, Epstein–Barr virus infection, smoking or childhood obesity, among others [1].

Vitamin D has strong immunoregulatory effects, through interaction with its specific receptor, expressed by almost all immune cell types, including macrophages/monocytes, neutrophils, T and B cells and dendritic cells, influencing the rate of transcription of vitamin D responsive genes. The vitamin D receptor (VDR) acts as a ligand activated transcription factor that binds to vitamin D response elements (VDREs) in gene promoters. The proximal promoter region of HLA-DRB1 of those bearing *HLA-DRB1*15* allele includes a functional VDRE and it has been reported that VDR can bind this VDRE, supporting the epidemiological evidence implicating vitamin D in the MS risk [4].

In addition, vitamin D may also exert a neuroprotective, neurotrophic or remyelinating effect in the CNS [5]. Its deficiency is considered a risk factor for autoimmune diseases, including MS [6]. However, it is difficult to discriminate whether lower vitamin D levels are a cause or a consequence of MS, since MS itself could potentially lead to low vitamin D levels, due to fewer outdoor activities and less sun exposure, and also, dietary differences.

Vitamin D is studied by measuring the concentration of serum 25-hydroxyvitamin D (25(OH)D). Supporting that low 25(OH)D level is a MS risk factor, lower levels have been observed in patients with clinically isolated syndrome, not related to less sun exposure due to MS-associated disability [7]. Similarly, an inverse association has been reported of both past vitamin D intake or 25(OH)D levels and future risk of MS [6].

Vitamin D is produced in the skin from sun exposure due to ultraviolet radiation (UVR). The melanin of the skin acts as a photoprotective filter, reducing the penetration of all wavelengths. People’s skin tone varies due to differences in the distribution, quantity, size, and type of melanin. In this way, people with a naturally dark skin tone have natural sunlight protection and require, at least, three to five times longer exposure to synthesize the same amount of vitamin D, as a person with a pale skin tone [8].

Melanin determines skin type, observed as white, brown or black skin. The cutaneous synthesis of 25(OH)D is determined by a complex range of variables among which skin pigmentation highlights. Skin melanin absorbs the UVR that initiates vitamin D synthesis and hence more pigmented skin has a decreased vitamin D synthesis for a given sun exposure compared to less pigmented skin [8].

Pale skin has been associated with MS, both a higher risk [9,10] and earlier onset [11], and a significant increase in burns has also been reported in MS patients [12].

Smoking, even by passive exposure, has recently been clearly associated with a increased risk of MS [13]. A dose-dependent response has been found both in smoking duration and in the number of cigarettes smoked, independently contributing to the increased risk. Additionally, a statistically significant interaction between *HLA-DRB1*15* and smoking has been demonstrated, in way that MS risk is higher among smoker subjects compared to non-smokers [14]. The harmful effect of smoking decreased slowly after quitting, independently of the cumulative dose [15].

Our aim has been to analyze the relationship between different MS risk factors (*HLA-DRB1*15:01* allele, sun exposure, vitamin D level, skin tone, and smoking habit) in independent cohorts from Spain, from three different provinces, with important differences in latitude, and thus, significant differences in UVR and erythemal index (Figure 1).

## 2. Experimental Section

### 2.1. Study Design and Subjects

In this retrospective case-control study we included 149 MS patients and 147 controls coming from three different provinces in Spain: Tenerife (28°N16°O), Málaga (36°N4°O) and Madrid (40°N3°O). All subjects included were permanent residents in these provinces and they are followed in the National Health System of Spain of their respective regions.

For the study size we had considered that 15% of the controls, in the Caucasian population, present the *HLA-DRB1*15:01* [16]. We considered as statistically significant an odds ratio greater than or equal to 2.5 (the most powerful predisposing genetic factor to MS has an OR around 2.5). Accepting an alpha risk of 0.05 and a beta risk of 0.2 in a two-sided test, we needed 140 patients and 140 controls (we have anticipated a drop-out rate of 10%). 149 prevalent patients diagnosed according to the McDonald criteria [17] from the three MS units (49 from HU Nuestra Señora de la Candelaria, Tenerife; 50 from HRU de Málaga, Málaga and 50 from HC San Carlos, Madrid) were recruited. Each patient had to give their informed consent and be accompanied by a “friend/control” of the same age (±3 years) (n = 147; 50 from HU Nuestra Señora de la Candelaria, Tenerife; 50 from HRU de Málaga, Málaga and 47 from HC San Carlos, Madrid) to be included in the study as a control. There were not familiar relationships between the patients themselves nor with the controls. None of the subjects were taking vitamin D supplements or were suffering from any disease known to influence vitamin D metabolism for the last six months prior to sampling. Demographic and MS clinical variables are shown in Appendix A.

All participants provided a blood sample for DNA and serum. All samples were taken from May to October and processed identically and following standardized procedures, immediately after their reception. DNA and serum aliquots were stored at −80 °C at the Malaga Regional Hospital Biobank, a node of the Andalusian Public Health System Biobank. Protocols were approved by the Institutional Research Ethics Committee (Comisión de Ética de la Investigación Provincial de Málaga (PI-0214-2014)). All experiments were performed in accordance with relevant guidelines and regulations.

Life-style factors, including smoking habits and sun exposure, as well as neurological data were collected by self-reported questionnaires and face-to-face interviews. Skin phototypes were evaluated by a unique research officer (Neurologist) in each province following the Fitzpatrick classification [18].

The first demyelinating neurological event, interpreted as the first MS relapse by a neurologist, was considered as the MS onset. The year when this occurred was defined as the index year for patients. The patients were matched with the controls and the index year for each control was the same as for the paired patient.

### 2.2. Serum Concentration of 25-Hydroxy Vitamin D

Serum concentration of 25-hydroxy vitamin D (25(OH)D) was measured by chemiluminescence immunoassay (CLIA) (LIAISON^®^ 25 OH Vitamin D TOTAL, Diasorin, Saluggia, Italy) in a Reference Laboratory (Barcelona, Spain).

We have corrected 25(OH)D levels for seasonality, sex and age.

The 25(OH)D level was modelled as a categorical variable in “normal” or “deficient”. The normal levels were considered above 20 ng/mL (50 nmol/L) [19].

### 2.3. Skin Phototype Classification

The Fitzpatrick skin type classification indicates the tolerance of the skin to UVR based on standardized color photographs and individual verbal response, sunburn, and tanning capacity, regarding first moderate unprotected sun exposure for a period of 45 to 60 min. Thereby skin types are classified as: Type I = pale white skin, always burns, never tans; Type II = white skin, always burns and sometimes tans; Type III = light brown skin, sometimes burns and always tans; Type IV = moderate brown skin, rarely burns and always tans; Type V = dark brown skin, never burns and always tans; and Type VI = deeply pigmented dark brown to black skin, never burns and always tans [18]. It is a reliable method irrespective of the month.

The skin tone was recorded on an ordinal scale and then categorized in “pale skin” (PS) = Type I + Type II and “light brown skin” (LBS) = Type III + Type IV. In our study population there were no individuals with skin type V or VI.

### 2.4. Exposure to Sun

In the personal interview, all subjects were asked for how long did they expose daily to sun, categorizing individuals into those who take “at least 30 min of sun every day” or “less than 30 min a day” (as recommended by the World Health Organization).

### 2.5. Smoking Habit

Smoking habit was considered positive when individuals smoked during or up to 10 years before the index year, as it has been shown that MS risk is decreased with length of time being smoking free, with no elevated risk for those with 10 or more years of being smoking free [15]. We have categorized subjects as “Smoker” or “Non-smoker”.

### 2.6. HLA Typing

The genotyping of HLA-DRB1 was performed using a low- and high-resolution allele-specific PCR amplification method (Olerup SSP^®^). Low-resolution genotypes were obtained by a combination of 24 PCR reactions, and for high-resolution genotypes 48 PCR reactions were performed additionally. As a positive control in each reaction, a second non-polymorphic genomic segment was amplified. Amplified products were separated by electrophoresis and visualized using ultraviolet illumination. We have categorized this variable in “Presence” or “Absence” of *HLA-DRB1*15:01*.

### 2.7. Statistical Analysis

The assessment of qualitative variables was performed by means of Chi-square tests.

We applied logistic regression (LR) to calculate the odds ratio with 95% confidence interval for the risk of developing MS. LR was used to associate the dependent variable “control/patient”, with the independent variables: presence of the risk allele, smoking and/or skin tone. The models were adjusted for province of residence and/or sex (we do not adjust models for age because there was no significant association in any model).

As data can be interpreted as clusters into three provinces/latitudes (Tenerife, Malaga and Madrid), we have used mixed effects logistic regression (with random intercept for each cluster).

To correct the 25(OH)D levels for seasonality, sex and age, we have introduced in our model the residuals obtained from the model that included the periodic function −sin (2πX/12) −cos (2πX/12), where X is number of month sample collection (not rounded) elapsed since the beginning of the year, age at sample collection and sex. The laboratory assay batch was not included because it is the same for all samples (they were processed at the same time at the Reference Laboratory) [20].

Receiver operating characteristics (ROC) generated from the LR analysis were performed and quantified using the area under the curve (AUC) to evaluate the predictive value of the different models. ROC curve analysis was performed with the data of probability predicted by the LR models for each individual of the sample in each of the models.

All tests were two-sided, and *p* < 0.05 was considered to be statistically significant. All analyses were performed using SPSS 15.0 statistical software (IBM, New York, USA), except for the mixed effects logistic regression model that was performed using the library lme4 [21] from R statistical package (Auckland, New Zealand) [22], version 3.6. The performance measure AUC has been estimated with correction for optimism using bootstrap resampling with the libraries pROC [23] and boot [24].

## 3. Results

### 3.1. Vitamin D and Skin Tone

Deficient levels of 25(OH)D were found in 60 out of 149 MS patients, whereas only 31 out of 147 controls showed this deficiency (65.9% vs. 34.1% respectively) (Table 1).

Vitamin D levels were finally not included in the LR model performed to evaluate the risk of developing MS, since they referred to the current levels, and therefore, could not be considered as a risk factor. For this reason, the variable included in the LR model was skin tone, because it is a variable easy to obtain which does not undergo changes over time.

According to this, in our study 92.3% (*n* = 84) of individuals with deficient levels of 25(OH)D had LBS, and only 14.6% (*n* = 7) of individuals with PS had deficiency levels of 25(OH)D (*p* = 0.007), although the individuals with LBS took more sun, observing that only 24.4% (*n* = 11) of PS individuals took at least 30 min of sun per day compared to 48.3% (*n* = 113) of those with LBS (*p* = 0.003).

We carried out a LR analysis to confirm the association among these three variables. Skin tone was the dependent variable (0 = PS; 1 = LBS), 25(OH)D levels and minutes of sun exposure per day were the independent variables (reference category: deficient levels of 25(OH)D and at least 30 min of sun exposure per day). This model showed that LBS subjects took significantly more sun than people with PS (OR = 2.97, IC95% = 1.42–6.23, *p* = 0.004) and that they had significantly deficient levels of 25(OH)D (OR = 5.07, IC95% = 1.74–14.77, *p* = 0.003) (model adjusted for seasonality for 25(OH)D).

To evaluate the discriminating power of this model, we obtained the AUC (AUC = 0.72 (0.64–0.80); *p* = 2.8 × 10^−6^), showing that 25(OH)D and sun exposure were associated with skin tone.

The group of patients (*n* = 149) showed a higher number of individuals with LBS compared to the controls (*n* = 147). There were no significant differences in the distribution by sexes (Table 1). However, there were differences in the distribution between the different latitudes, with significantly more people with LBS in Tenerife and Málaga (92.9% (*n* = 92) and 96.0% (*n* = 96) respectively) than in Madrid (61.1% (*n* = 58)) (*p* = 3.23 × 10^−12^).

The LR analysis to evaluate the relationship between skin tone and the risk of developing MS, adjusted by sex, showed that individuals with LBS had 4.4-fold more MS risk than those with PS (model 1, Table 2).

Besides, a moderate negative association between skin tone and province of residence was found (gamma = −0.70, *p* = 6.2 × 10^−8^; R de Pearson = −0.35, *p* = 9.3 × 10^−10^). Due to this multicollinearity, the province of residence was not included as a variable in the LR model.

### 3.2. Smoking Habits

Out of 149 patients, 103 (69.1%) reported to be smokers at MS onset or to have smoked in a period of ten years before the onset. However, out of 147 controls, only 51 (34.7%) reported it. This significant differential distribution of smoking habits between MS patients and controls was observed in two of the three provinces (Table 1).

There were not significant differences in the distribution of smokers by sex in any of the groups (Table 1).

The LR analysis showed that smokers had 4.4 times more MS risk than non-smokers, in a model adjusted for sex and province of residence (model 2 and Table 2).

### 3.3. HLA Typing

As expected, the group of patients (*n* = 149) showed a significantly higher percentage of individuals with the *HLA-DRB1*15:01* allele than control group (*n* = 147) (Table 1). There were no significant differences in the distribution of this allele by sexes (Table 1) or by province of residence (Tenerife: 23.2% (*n* = 23), Málaga: 26.5% (*n* = 26) and Madrid: 23.7% (*n* = 23)).

The LR analysis showed that the presence of the *HLA-DRB1*15:01* allele significantly increased the MS risk with an OR = 2.3 in a model adjusted by sex and province of residence (model 3 and Table 2).

### 3.4. Multifactorial Model

To study the association of skin tone, smoking habits and presence of *HLA-DRB1*15:01* with the susceptibility to MS we performed a mixed-effect logistic regression. Carrying out this model the log odds of the outcomes were modelled as a linear combination of the predictor variables and data were clustered into provinces (Tenerife, Malaga and Madrid).

Our model, that included 149 patients and 147 controls, showed (after adjusting by sex) that females with LBS, smokers and those who had *HLA-DRB1*15:01* allele had a higher MS risk versus males with PS, non-smokers and who did not have the *HLA-DRB1*15:01* allele. In addition, LBS tone was shown as the strongest risk factor for developing MS, followed by smoking, and having the presence of the *HLA-DRB1*15:01* allele (model 4 and Table 2).

We compared the discriminating power of the different models to assess the risk of developing MS. Models 1 (skin tone and sex) and 3 (*HLA-DRB1*15:01*, sex and province of residence) do not have sufficient discriminatory power. Model 2 (smoking habit and sex and province of residence) showed higher discriminative power with an AUC of 0.70 (0.64–0.76) (*p* = 2.74 × 10^−9^). But the highest predictive value was found with the multifactorial model 4 (skin tone, smoking habit, *HLA-DRB1*15:01* and sex), showing an AUC of 0.748 (0.69–0.80) (*p* = 2.27 × 10^−13^) (Figure 2). The AUC we got with this model had a very similar bootstrap optimism-corrected ROC area of 0.765 (0.69–0.82) that indicates that there is no sign of overfitting.

### 3.5. Accumulative Risk Model

When we examined the combination of the different factors, we observed that only 4.8% of the subjects without any risk factors were MS patients. Of the individuals who had LBS as the only risk factor, 33% were patients. The percentage of MS patients increased in the group of subjects with LBS plus the presence of *HLA-DRB1*15:01* (56.0%), was higher when the two factors were LBS plus smoking (66.7%) and became maximum when we considered the presence of the 3 factors, LBS, smoker and presence of *HLA-DRB1*15:01* (83.3%) (*p* < 0.00001) (Figure 3).

To examine how these combinations affected the increase in risk, an accumulative LR model was carried out that showed that subjects with only LBS had 10.5-fold more risk of MS than subjects without any risk factor. Those people with LBS plus presence of *HLA-DRB1*15:01* had 2.5-fold more risk than people with LBS alone. But if the combination of two factors was having LBS plus smoking, the risk was increased 4-fold compared to people with LBS alone. And when the genetic factor was added to the latter combination the risk was increased by 2.6-fold (Table 3). This model has a high predictive value (AUC= 0.76 (0.70–0.81), *p* = 5.76 × 10^−13^).

## 4. Discussion

In this study we analyze the relationship of different factors with the risk of developing MS. We analyzed three populations from provinces with 12, 8 and 4 degrees of North latitude difference (Tenerife, Malaga, Madrid, Spain), in a single country (Spain), with Caucasian populations that were very similar genetically [25], but with subtle differences (Figure 1).

The multifactorial LR analysis showed that females with LBS, who were smokers and have the *HLA-DRB1*15:01* allele had an increased risk of developing MS versus males with PS, who were non-smokers and who did not have the *HLA-DRB1*15:01* allele.

According to our model, the greatest risk of developing MS is provided by the skin tone. Subjects with LBS (type III + type IV) had significantly 4.6 times more MS risk than those with PS (type I + type II). It is very uncommon to find individuals with skin type V or VI in our MS population because the Spanish population is very homogeneous and until approximately the last two decades, we did not have significant immigration. Immigrants currently represent 9.9% of the Spanish population and mainly come from South America and Morocco (white or mestizo population) [26]. In hospitals of the National Health System of Spain, where MS patients are followed-up and treated, it is very rare to find black patients, and MS is almost unknown in that segment of population.

Historically, from old epidemiological studies, it has been accepted that blacks have a lower MS risk than whites. Usually it is admitted that MS risk given ancestry is highest in Northern Europeans, then admixed population, and is near absent in the black population of Africa. Northern European ancestry is also associated with pale skin tone and low vitamin D levels, and a higher frequency of *HLA-DRB1:15:01*. In this line, it seems surprising that people with LBS, in our study, have more risk of developing MS than people with PS. This could be related to a persistent bias, due to the lack or insufficient access of this population to neurological assistance, not only in the past, but even today, as a smaller proportion of blacks are treated or evaluated at MS centers, clinics or neurologist practices compared to whites [27]. However, since 2011 several authors have shown a similar or even higher incidence of MS in blacks than in whites, and it has been demonstrated that blacks are diagnosed, in general, at a later age and their disease course is more aggressive compared to the Caucasians, although these conclusions remain to be confirmed, as the socioeconomic environments are very different between these two populations in every reported country up to now [28,29,30,31].

Moreover, the skin tone has been related with vitamin D levels, with a higher prevalence of vitamin D deficiency reported in individuals with darker skin tones [8], this was also confirmed in our population, where subjects with LBS had significantly lower vitamin D levels than those with PS. It has been reported that these individuals need more UVR and thus, more sun exposure, to equal vitamin D levels than people with PS [8]. Another possible explanation of the association of low vitamin D levels with darker skin tones could be due to genetic factors related to changes in the affinity of vitamin D binding protein for vitamin D. It has been described that genetic determinants that are related to vitamin D levels may be race specific [32] and our population is genetically homogeneous, with all individuals of the same ethnicity.

Although a generalized vitamin D insufficiency in the general population has been reported [33], a fact that we have confirmed, we have observed, a significantly higher percentage of subjects with vitamin D deficiency in MS patients compared to the controls in accordance with results previously reported for early stages [34] as well as for later stages of the disease [35]. The role of vitamin D in MS can be related to its immunomodulatory effects, as it has been shown that this vitamin is able to decrease Th1 activity and increase Th2 and T-regulatory cell activity [36], and its deficiency could be partly explained by abnormalities in the vitamin D metabolism pre-existing in MS patients [37].

The body mass index (BMI) is another factor that has been correlated with low levels of vitamin D [38]. We could not include it as a factor in our risk model because we do not have this data prior to the disease onset.

The second factor that contributes to the MS risk in our model was smoking, known as one of the most established environmental risk factors related to MS, not only in its susceptibility but also in its clinical severity [13]. Because the patients included in this study were matched by age with the controls and they had relatively short disease duration, we believe that there was no temporal difference between them.

Exposure to tobacco smoke has been related to the development of many diseases [39]. Smoking is implicated in the production of many cytokines, enhancing Th1 and Th17 polarization, having far-reaching effects on chronic inflammation and autoimmunity at a systemic level [40]. It has been related with several autoimmune diseases by regulating response of Th17 cells. Moreover, it can alter the lung that could contribute to the activation of potentially auto-aggressive T cells, and its subsequent migration to the target tissue in MS [41].

We must also consider other possible adverse effects of tobacco linked to MS, as chronic cyanide intoxication leading to demyelination and increased frequency and persistence of smoking-mediated infections [42]. In addition, long-term exposure to tobacco produces an increase in the permeability of the blood brain barrier (BBB), which in some way contributes to increasing brain damage [43].

The last factor in our model is the presence of the *HLA-DRB1*15:01* allele. This genetic risk factor was established more than 40 years ago [44]. Epidemiological studies have shown that the *DRB1*15:01* allele is the HLA allele most closely associated with MS susceptibility in Caucasian populations [45].

There are limitations to this study. Although the number of individuals included allows us to achieve sufficient statistical power, a greater sample size is warranted. Another limitation is that it is a retrospective study. Furthermore, there is the risk that “friend controls” could not be representative of the general population.

## 5. Conclusions

As a conclusion, our study indicates that MS risk is determined by complex interactions among sex, environmental factors, and genotype. We suggest that in these complex interactions the milieu influence could play a very important role in the development of the disease, probably by being able to activate the appropriate cell populations, to affect cytokine production and/or to cause direct effects on the BBB. We propose that the proinflammatory environment due to low levels of vitamin D (explained in our model by the skin tone) and tobacco could drive an autoimmune attack against myelin in the CNS by self-reactive lymphocytes, whose entrance into the CNS would be favored by the effect of tobacco on the BBB. If this could be definitively confirmed, some preventive strategies might be envisaged to address the increased incidence of MS, and to try to avoid what is starting to be considered as an epidemic [46,47].

## Figures and Tables

**Figure 1 jcm-09-02384-f001:**
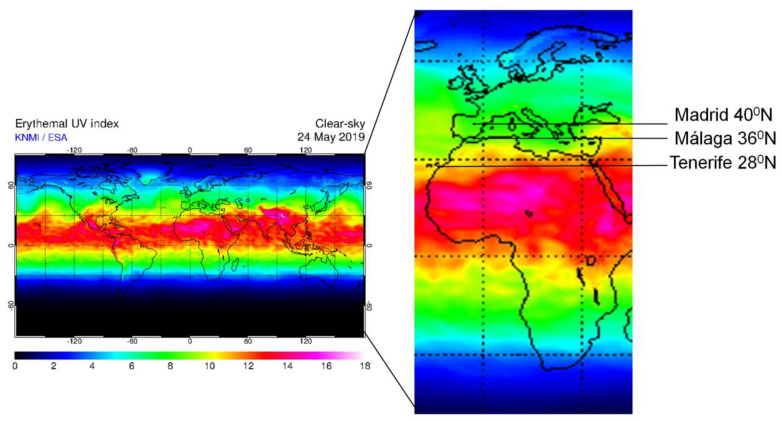
Map of world ultraviolet radiation erythemal index, with the three provinces reported in this study. This index is a measure of ultraviolet radiation (UVR) at ground level on the Earth, being an estimation of the UV levels that have important effects on human skin. Modified from: http://www.temis.nl/uvradiation/UVI/uvief0_wd.gif (accessed 24th of May 2019).

**Figure 2 jcm-09-02384-f002:**
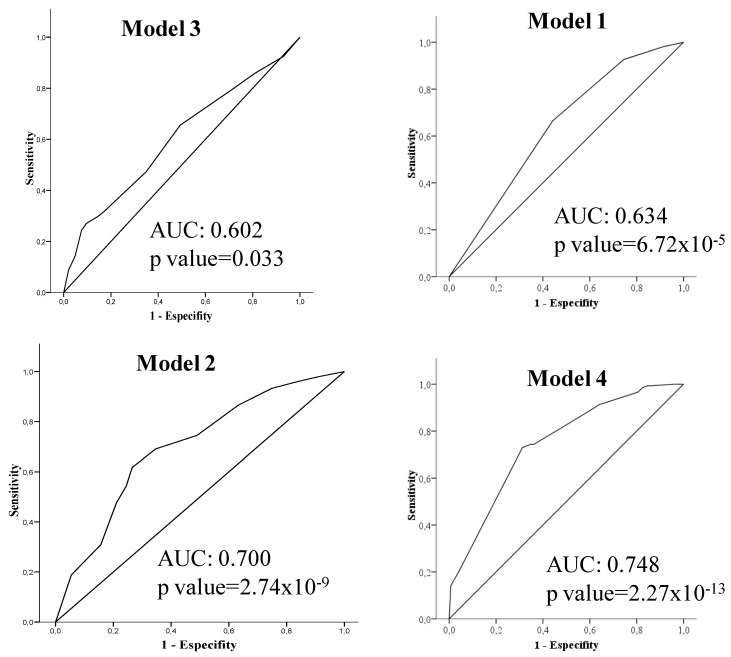
Receiver operating characteristics (ROC) curves generated from the logistic regression analysis to evaluate the models of MS risk. The discriminating power (the area under the ROC curve (AUC)) of the models performed was increased in the following order: model 3 (independent variable: *HLA-DRB1*15:01*), model 1 (independent variable: skin tone) and model 2 (independent variable: smoking habit). The predictive capacity was the highest in the multifactorial model 4 (independent variables: *HLA-DRB1*15:01*, skin tone, smoking habit).

**Figure 3 jcm-09-02384-f003:**
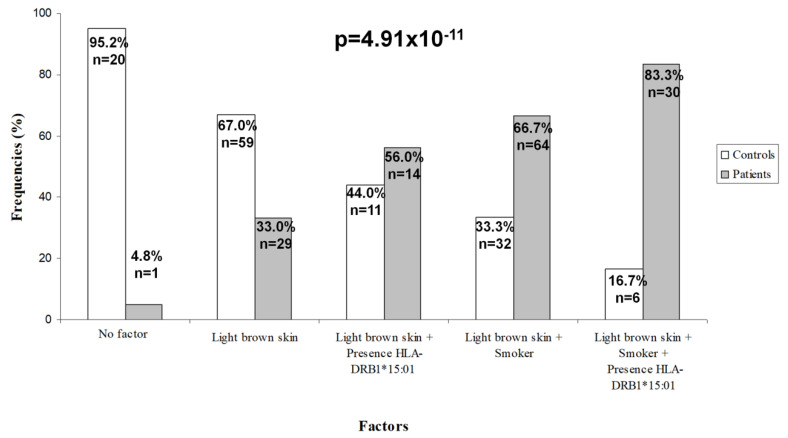
Frequencies for accumulative risk model. Frequencies for each group according to the combination of different risk factors.

**Table 1 jcm-09-02384-t001:** Relative (%) and absolute (n) frequency of study variables.

	Women% (n)	Men% (n)	*p* Value	Patients% (n)	Controls% (n)	*p* Value
25(OH)D deficient	33.0% (65)	26.3% (26)	n.s.	40.3% (60)	21.1% (31)	3.50 × 10^−4^
Light brown skin	83.6% (163)	83.8% (83)	n.s.	92.6% (138)	74.5% (108)	2.60 × 10^−5^
Smokers	51.0% (100)	54.5% (54)	n.s.	69.1% (103)	34.9% (51)	4.13 × 10^−9^
Smokers in Tenerife	53.3% (32)	56.4% (22)	n.s.	73.5% (36)	36.0% (18)	1.81 × 10^−4^
Smokers in Málaga	45.8% (33)	64.3% (18)	n.s.	58.0% (29)	44.0% (22)	n.s.
Smokers in Madrid	54.7% (35)	43.8% (14)	n.s.	76.0% (38)	23.9% (11)	3.39 × 10^−6^
Present *HLA-DRB1*15:01*	24.1% (47)	25.3% (25)	n.s.	31.8% (47)	17.1% (25)	0.004

**Table 2 jcm-09-02384-t002:** Association of different variables with multiple sclerosis (MS) risk.

	OR	IC 95%	*p* Value
**Model 1**			
Skin tone (Light brown skin vs. pale skin)	4.40	2.13–9.06	5.34 × 10^−5^
Sex (female vs. male)	1.71	1.03–2.83	0.037
**Model 2**			
Smoking habits (Smoker vs. non−smoker)	4.39	2.67–7.21	4.96 × 10^−9^
Sex (female vs. male)	1.85	1.09–3.12	0.023
Province of residence			
Tenerife vs. Madrid	0.87	0.47–1.60	0.66
Málaga vs. Madrid	0.88	0.48–1.61	0.68
**Model 3**			
*HLA−DRB1*15:01* (Presence vs. absence)	2.30	1.32–4.02	0.003
Sex (female vs. male)	1.66	1.01–2.74	0.045
Province of residence			
Tenerife vs. Madrid	0.95	0.54–1.69	0.95
Málaga vs. Madrid	0.89	0.50–1.59	0.70
**Model 4**			
Skin tone (Light brown skin vs. pale skin)	5.90	2.39–15.45	1.72 × 10^−4^
Smoking habits (Smoker vs. non−smoker)	4.52	2.69–7.72	1.89 × 10^−8^
*HLA−DRB1*15:01* (Presence vs. absence)	2.39	1.30–4.50	0.006
Sex (female vs. male)	1.88	1.08–3.30	0.025

Dependent variable: 0: control; 1: MS patients. Model 4: mixed effects logistic regression.

**Table 3 jcm-09-02384-t003:** Accumulative risk model.

	OR	IC 95%	*p* Value
Number of factors			
Light brown skin vs. 0 factor	10.51	1.34–82.65	0.25
Light brown skin + Presence *HLA−DRB1*15:01* vs. 0 factor	26.33	3.02–229.35	0.003
Light brown skin + Smoker vs. 0 factor	42.68	5.45–334.43	3.52 × 10^−4^
Light brown skin + Smoker + Presence *HLA−DRB1*15:01* vs. 0 factor	114.16	12.60–1034.12	2.51 × 10^−5^
Sex (female vs. male)	1.89	1.06–3.37	0.031

Dependent variable: 0: control; 1: MS patients.

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
