# Peer review of "Skin Phototype Could Be a Risk Factor for Multiple Sclerosis"

_jcm, 2020, doi:10.3390/jcm9082384_

Round 1

Reviewer 1 Report

The manuscript entititled "Skin phototype could be a risk factor for multiple sclerosis" is interesting. I appreciate the attempt at synthesizing a set of risk factors together in one study. There are some issues I have: 

-This is a retrospective study and should be mentioned in discussion as a study limitation

-It is not clear to this reader that assessment of skin tone from May to October is reliable due to tanning. Please comment.

-It is not clear that vitamin D levels were corrected for seasonality. See Munger et al. JAMA 2006 296:2832

-It is not stated whether patients were taking Vit D supplements or not

-Vitamin D binding protein's affinity for Vit D varies between individuals due to genetic factors. This is an another possible explanation of the association of low Vit D with increased skin pigmentation, rather than sun exposure. Differences in Vit D may also be a product of dietary differences, which may associate with culture/ethnicity.

-Body mass index is associated with low vitamin D, and this may be a product of increased fat storing more vitamin D away from blood stream. This should be addressed if possible in analysis.

-It is not mentioned whether the subjects of this study were always living at the study sites for their whole lives or whether they emigrated from other regions. If subjects emigrated from more northerly regions, this may increase MS risk, and also have effects on Vit D levels depending on timing.

-The manuscript may benefit from some shortening/focus.

Reviewer 2 Report

Abstract: “we have not only analyzed the impact of… best known factors… but we have included a new one (skin phototype) that has not been analyzed so far”. This is not true. Melanin density (skin phenotype) was studied in a landmark 2003 paper by van der Mei (Pubmed ID [PMID]: 12907484), and a few others: PMID: 21300969, PMID: 18711112, and including studies that indirectly assess skin phenotype by studying sunburning (i.e. PMID: 20463038). Please review the literature accordingly and revise the manuscript, and clearly articulate what is new about the current study.

Pg3 line 87. It is unclear why such a large odds ratio (2.5) was used in the power calculation. Can you generate post-hoc power based on sample size – this will guide the interpretation.

Pg3 line 89. Were these incident MS cases? Or prevalent cases? Demographics on the cases and controls are presented nowhere in the document. Please add, including onset age, disease duration, education (as a proxy for wealth – also a potential risk factor), body mass index (which is a key risk factor for MS), etc. This is major concern because if cases are prevalent (having had disease for many years), then the index year for the questions (i.e. smoking status) temporal differs between cases and controls. This is a major factor for the smoking relationship as we know global trends in using combustible tobacco products have steadily decreased the last twenty years. So compare data from controls at interview versus cases from onset a few years prior might introduce significant information bias. Appropriate considerations in the models are necessary if this is the case.

Pg3 line 92. Were the friend/control unrelated to cases? Were all cases also unrelated?

Pg4 line43. Given the modest sample size and absence of replication data set, do add cross-validation to estimating the AUC and/or bootstrapping to generate bias-corrected confidence intervals. This is essential since you are constructing risk prediction models.

Table 1 – please label HLA-DRB1*15:01 as present/absent

Pg9 line 261. “People of African origin have lower MS risk…. Etc” the authors are confusing ancestry with origins. This entire paragraph needs to be rephrased in the context of MS risk given ancestry with highest risk in Northern Europeans, then admixed population, and near absent in Africa.

The most significant concern which was not discussed or statistically considered in any way was ancestry. Northern European ancestry confers the greatest risk for MS, and it is also associated with skin tone and vitamin D levels, and the frequency of HLA-DRB1:15:01, thus this is a major confounder that was not accounted for in any analysis or anywhere in the discussion, possibly biasing most of the associations. This is a significant and critical omission that makes drawing any conclusions near impossible.

Reviewer 3 Report

Thank you for the ability to review this interesting manuscript, which describes adequately the independent role of skin phototype as risk factor for MS. Below you can find my comments- suggestions for improvement:

  1. Introduction: In order to give a more thorough background on the role of already known risk factors, you could add a brief comment on the combined role of HLA-DRB1*15 and Vitamin D, via the VDRE in the promoter region of the gene (Ramagopalan et al. PLoS Genet 2009), as well as a brief comment on the combined role of HLA-DRB1*15 and smoking (RR increase X13, Hedstrom et al. Brain 2011).
  2. Study design and subjects: Please clarify if the 'research officer' who evaluated the skin phototypes was a Dermatologist, or had other special training on the matter.
  3. Results: Rows 163-167, 'Melanin - skin [7]'. I would prefer to read this in the Introduction, or even the Discussion section.
  4. Results: Please explain why you had missing values on the principal variable 'skin phototype' (149 patients instead of 150, 145 controls instead of 150).
  5. Results - 3.5 Accumulative risk model: I would like to know the percentages of patients, when the only risk factor was HLA-DRB1*15:01, and smoking, accordingly.
  6. Discussion: Rows 266-267. 'But since 2011 several authors have shown a similar or even higher incidence of MS in people of African origin..'. This is indeed a really interesting information, backed up by the findings of the study. I would like to see additional literature on this. My impression is that ref.16 mainly discusses severity and not incidence of disease among ethnic groups.
  7. Minor word editing comments: Row 106 - was considered as the MS onset / Row 125: Exposure to sun / Row 161: because it is a variable / Row 162: omit 'and'.

Reviewer 4 Report

Your paper raises several interesting points that the authors (all investigators from Spain) should consider to be followed with further studies. Interestingly, your study population involving areas in central, southern Spain and an island off the coast of northwest Africa, case recruitment did not find subjects with Fitzpatrick skin types V and VI. It is known that most of the African populations residing in Spain with dark brown (V) and black skin (VI) are a minority generally formed by immigrants. This reviewer believes the prevalence of MS would be very low, minimal or non-existent in these population groups. This observation probably would be different in other areas, i.e. USA, the Caribbean, Brazil, etc. I believe a comment in this respect would be reasonable and clarifying.   

Round 2

Reviewer 2 Report

The written language of this manuscript can be improved in several instances; for example, line 47: “Besides, it has been demonstrated the relationship with MS pathogenesis of several environmental factors such as high latitude, vitamin D deficiency, Epstein–Barr virus infection, smoking or childhood obesity, among others”

Line 46: 233 common MS risk variants. Also HLA-DRB1*15:01 should be italicized throughout – as is standard for gene names.

Please replace the racist language of “clear skin” and “clean skin” (CS) to fair or pale skin tone. Here are examples of more appropriate scientific language for describing the Fitzpatrick scale: https://www.arpansa.gov.au/sites/default/files/legacy/pubs/RadiationProtection/FitzpatrickSkinType.pdf, https://www.facebook.com/IOSHofficial/photos/a.10150196833679010/10155641602059010/?type=3&theater, https://www.picuki.com/media/2248522272401716124

Line 102: The power calculation remains unclear. Did the authors use the frequency of smoking alongside the measure of association for HLA-DRB1*15:01 to determine sample size? Why not use the expected frequency of HLA-DRB1*15:01 for a Spanish population?

Table S1. It is not state whether it is standard deviations or standard errors in the table.

Line 122: Lifestyle factors where collected by survey. In line 126 it seems that index year for the survey for a case was onset year while for a control it was year of participation (possibly 2018?). If this is the study design, then this case-control study is confounded by time. The authors state that the average disease duration for cases of 11.5 year +/- 9.0 is a short disease duration, I disagree. Based on the standard errors, several cases have a disease duration of at least 20 years, thus temporal trends in smoking and sun exposure behaviors would be severely different than controls who were interviewed in ~2019. Furthermore, these cases with 10-20 years of disease duration are being asked about their past behaviors at ages 20-30 years when MS presented itself, while controls are being asked about their current exposure profiles at age 40 (age at interview). Therefore, it is not surprising the differences in smoking status is so dramatic between cases and controls – it is being driven by very difference exposure periods, both in relationship to calendar time (therefore different temporal trends and the potential for recall bias) and chronological age (different behaviors due to age in the index year). This is also a concern for all sun exposure analyses. Time/age was not appropriately considered in all models, therefore all reported associations could be severely biased. Unfortunately, this survey data likely introduces bias versus adding useful information.

Line 152. How was being a smoker defined (e.g. smoked 100 tobacco products over their lifespan, or within a month, or some other measure)?

Line 155: MS risk is not reversed after a decade of smoking cessation. Among former smokers risk for MS decreased with length of time being smoking free, with not elevated risk for those with 10 or more years of being smoking free.

Line 169. Because you are asking cases and controls to report their exposures with respect to different index years, you should include age during the index year.

Line 216. Instead of including province as a covariate, the authors should consider adjusting the standard errors for province (i.e. clustering by province).

Line 345. Limitations need to be expanded to address several issues, including the fact that friend controls are likely not representative of the general population.
